# Participation Restrictions among Children and Young Adults with Acquired Brain Injury in a Pediatric Outpatient Rehabilitation Cohort: The Patients’ and Parents’ Perspective

**DOI:** 10.3390/ijerph18041625

**Published:** 2021-02-08

**Authors:** Florian Allonsius, Arend de Kloet, Gary Bedell, Frederike van Markus-Doornbosch, Stefanie Rosema, Jorit Meesters, Thea Vliet Vlieland, Menno van der Holst

**Affiliations:** 1Basalt Rehabilitation Center, Department of Innovation, Quality and Research, 2543 SW The Hague, The Netherlands; A.deKloet@basaltrevalidatie.nl (A.d.K.); F.vanMarkus@basaltrevalidatie.nl (F.v.M.-D.); J.Meesters@basaltrevalidatie.nl (J.M.); t.p.m.vliet_vlieland@lumc.nl (T.V.V.); 2Department of Occupational Therapy, Tufts University, Medford, MA 02155, USA; gary.bedell@tufts.edu; 3National Department Level, Specialists in Youth and Families, 1105 AZ Amsterdam, The Netherlands; stefanie.rosema@gmail.com; 4Department of Orthopaedics, Rehabilitation and Physical Therapy, Leiden University Medical Center, 2333 ZA Leiden, The Netherlands; 5Centre of Expertsie in Health Innovations, The Hague University of Applied Sciences, 2521 EN The Hague, The Netherlands

**Keywords:** participation, rehabilitation, acquired brain injury, pediatric, patient-report, parent-report

## Abstract

Improving participation is an important aim in outpatient rehabilitation treatment. Knowledge regarding participation restrictions in children and young adults with acquired brain injury (ABI) is scarce and little is known regarding the differences in perspectives between patients and parents in the outpatient rehabilitation setting. The aims are to describe participation restrictions among children/young adults (5–24 years) with ABI and investigating differences between patients’ and parents’ perspectives. At admission in 10 rehabilitation centers, patients and parents were asked to complete the Child and Adolescent Scale of Participation (CASP; score 0–100; lower score = more restrictions) and injury/patient/family-related questions. CASP scores were categorized (full/somewhat-limited/limited/very-limited participation). Patient/parent-reported outcomes were compared using the Wilcoxon signed-rank test. 223 patients and 245 parents participated (209 paired-samples). Median patients’ age was 14 years (IQR; 11–16), 135 were female (52%), 195 had traumatic brain injury (75%). The median CASP score reported by patients was 82.5 (IQR: 67.5–90) and by parents 91.3 (IQR: 80.0–97.5) (difference = *p* < 0.05). The score of 58 patients (26%) and 25 parents (10%) was classified as ‘very-limited’. Twenty-six percent of children and young adults referred for rehabilitation after ABI had “very-limited” participation. Overall, parents rated their child’s participation better than patients themselves. Quantifying participation restrictions after ABI and considering both perspectives is important for outpatient rehabilitation treatment.

## 1. Introduction

Acquired brain injury (ABI) refers to irreversible damage to the brain which either has a traumatic cause; i.e., caused by external trauma (TBI) or a non-traumatic cause (nTBI); i.e., by internal causes [1]. It is a common diagnosis in children and young adults. The estimated yearly incidence rates in the Netherlands per 100,000 children and young adults are 288.9 (0–14 years) and 296.6 (15–24 years) for TBI and 108.8 (0–14 years) and 81.5 (15–24 years) for nTBI, respectively [2]. Due to natural brain adaptation, the majority of children and young adults with ABI will recover within the first year after brain injury [3]. However, on average, approximately 30% have persisting problems, and this group may benefit from rehabilitation treatment [1,2,3,4,5]. One of the ultimate goals of (outpatient) rehabilitation treatment is optimizing a patient’s daily life participation [2,6,7,8,9,10]. However, despite its relevance, knowledge on participation restrictions of children and young adults with ABI referred for rehabilitation treatment is scarce. The currently available literature focuses on children (<14 years) with TBI in hospital-based cohorts [10,11,12,13,14,15,16,17,18].

Only a few studies focus on both patients’ and parents’ perspectives, and knowledge regarding outcomes on participation measuring both perspectives is even more scarce [9,12,14,19,20]. Moreover, for the pediatric rehabilitation-based population, and in the context of family-centered care, the question is whether the severity and nature of participation restrictions can best be rated by patients, parents or both, which is still an under-researched area [20,21,22,23,24].

Two relevant studies (a study in the United States (US) and a Dutch study) found strong internal structure validity and internal consistency between the patient and parent reported versions of the outcome measures i.e., the Child and Adolescent Scale of Participation (CASP) [9,20]. Yet, discrepancies between patients’ and parents’ perspectives were found, where parents reported lower scores than the patients [9,20]. However, the study conducted in the US only focused on youth aged 11–17 years and with chronic conditions/disabilities, and making comparison to patients with ABI difficult [20]. The Dutch study focused on patients with ABI a small age range (14–25 years), and used a relatively small sample size (*n* = 49) from only one rehabilitation center [9]. This rehabilitation-based study in which the primary focus was on fatigue outcomes, investigated participation as well and found multidirectional relationships between participation and fatigue as well as considerable participation restrictions among patients with ABI as measured with the CASP (median 82.5, IQR 68.8, 92.3) [9].

Other studies based on hospital-based cohorts, report that 25–80% of children and young adults with either TBI (mild/moderate/severe) or nTBI (i.e., stroke, tumor) experience participation restrictions after ABI [2,6,7,9,10,14,16,17,25,26,27,28,29,30,31,32,33,34,35,36]. This wide range is due to differences in definition of participation, outcome measures, inclusion criteria (i.e., age, type and severity, hospital based) and time points (i.e., time since onset of ABI) used in these studies [36]. In both children and young adults, participation restrictions after ABI tend to persist for a long time which negatively influences life development [37]. Negative consequences could affect the development of physical, psychological and social emotional skills and competencies, as well as the shaping of identity, health and wellbeing in adulthood [2,7,9,16,17,25,30,31,32,33,34,35,36,38,39,40].

Regarding the factors associated with participation restrictions, several studies found that more participation restrictions after pediatric ABI were associated with (among others), diminished health-related quality of life (HRQoL), and negative patient and environmental influences i.e., more patient’s motor, cognitive, behavioral and emotional consequences [7,12,16,22,23,36,41,42]. To date, these influences were not investigated among children and young adults with ABI who were referred for outpatient rehabilitation treatment.

The present study aims to investigate among children and young adults with ABI (5–24 years with TBI or nTBI) who were referred for outpatient rehabilitation treatment (not having received any prior rehabilitation treatment):the nature and severity) of participation restrictions;differences regarding patients’ and parents’ perspectives on patients’ participation restrictions;the association between HRQoL and patient- and environmental factors on the one side and participation restrictions on the other side.

## 2. Materials and Methods

### 2.1. Design

Data from patients with ABI (and/or their parents) that were referred for outpatient rehabilitation treatment on the basis of continuing and/or expected problems, related to their brain injury were analyzed. These patients had not received any outpatient rehabilitation treatment yet. This study was part of a larger multi-center study on family impact, fatigue, participation and quality of life and associated factors in the Dutch ABI population (children and young adults). The study was started in 2015 in 10 Dutch rehabilitation centers, using a consensus-based set of patient/parent-reported outcome measures (PROMs) at admission as part of routine care. The reports of these PROMs were used for clinical goal setting in rehabilitation practice. The protocol for this study was reviewed by the medical ethics committee of the Leiden University Medical Center (P15.165), and an exempt from full medical ethical review was provided. For the current article the ‘Strengthening the Reporting of Observational studies in Epidemiology’ (STROBE) guidelines were used [43].

### 2.2. Patients

All children and young adults aged 5–24 years with a diagnosis of ABI, who were referred for outpatient rehabilitation treatment to a participating rehabilitation center and their parent(s) were eligible to participate. If patients and/or parents were unable/limited to write and/or understand the Dutch language, they were not invited by the center’s health care professionals to complete the questionnaires. Patients over the age of 16 years had to give their parents’ permission for completing the questionnaires according to the Dutch law of healthcare decision making.

### 2.3. Data Collection

Demographic and injury characteristics were extracted from the medical records by health professionals employed by the rehabilitation centers where patients had their appointment. For the outcomes related to participation, quality of life and child and environmental outcomes a (digital) questionnaire was administered to patients and/or their parents. Patients and parents were given the opportunity to complete this questionnaire prior to the first appointment during their visit at the outpatient rehabilitation clinic. If a patient (in case of a young adult) came without parents to the appointment, parents were asked to complete the questionnaires either on paper or digitally within one week after the first appointment. Unique links to the digital questionnaires were sent to the participants by e-mail by the medical health professionals working at the rehabilitation centers. Data were recoded, and thereafter anonymously stored in a central database at Basalt rehabilitation center in The Hague (The Netherlands). Finally, after analyzing the data, the centers received the results to use for clinical practice.

### 2.4. Assessments

#### 2.4.1. Demographic and Injury Characteristics

Information regarding demographics and injury-related characteristics included: date of birth, date of injury, date of referral to rehabilitation, age at the start of the first appointment i.e., the difference between date of birth and date of referral to rehabilitation and gender i.e., male/female. Time between onset of ABI and referral to rehabilitation was calculated and thereafter divided into 2 groups: referred for rehabilitation within 6 months, and after 6 months after ABI onset. The categorization of ABI was divided in: TBI/nTBI. If known, the TBI severity levels were divided into either mild, or moderate/severe (based on the Glasgow Coma Scale at hospital admission [44]). NTBI causes were divided into stroke/cerebrovascular accidents, brain tumors, meningitis/encephalitis, hypoxia/intoxication, and other.

#### 2.4.2. Participation Outcome Measure

The Child and Adolescent Scale of Participation (CASP) was administered to patients and parents to measure participation restrictions of the patient. The CASP is part of the “Child and Family Follow-up Survey” (CFFS) [45]. The CFFS, including CASP was validated for children, young adults and youth with ABI, was translated in the Dutch language, and is considered feasible and reliable tools to assess participation restrictions [2,17,20,25,45,46,47,48]. Patient-report (both children and young adults) and parent-report versions of the CASP were available and used both in the present study [17,20,47]. The CASP is a 20-item questionnaire, yielding a total score, and 4 domain scores including: home & community living activities; 5 items, home participation; 6 items, community participation; 4 items, and school/work participation; 5 items. Activities regarding participation are rated on a 4-point scale: 4 = age expected (full participation), 3 = somewhat limited, 2 = very limited, and 1 = unable. Items marked as” not applicable” do not receive a score. Scores for each item are summed and divided by the maximum possible score based on the number of items rated. The results, multiplied by 100, give a final score between 0–100, which counts for both the total score and the domain scores. The higher the scores, the closer a patient is participating to age-expected participation levels in daily life.

#### 2.4.3. Four-Level Categorization

For the present study, a 4-level categorization system was developed to distinguish between levels of participation restrictions of patients for use in clinical practice. First, a draft version of a 4-level categorization was created by five of the authors based on preliminary analysis of the CASP data gathered for the present study and consensus discussions (F.A., A.d.K., M.H., G.B. and T.V.V.). We thereafter presented the categorization to a group of physicians and psychologists in the field, and to the remaining authors who are all experts in the field. Together, consensus was reached on the categorization and it was agreed to use it for further analyses in the present study. The 4-level categorization was made as follows:-Category 1, CASP score 100–97.5: Full participation; participating in activities the same as or greater than peers, with or without assistive devices or equipment.-Category 2, CASP score 97.5–81.0: Somewhat limited participation; participating in activities a bit less than peers. The patient may also need occasional supervision or assistance.-Category 3, CASP score 81.0–68.5: Limited participation; participating in activities less than peers. The patient may also need supervision or assistance.-Category 4, CASP score 68.5 or less: Very limited participation; participating in activities much less than peers, the patient may also need a lot of supervision or assistance.

#### 2.4.4. Secondary Outcome Measures

When assessing participation restrictions, patient (i.e., children and young adults) factors, environmental factors as well as health related quality of life were described using the following outcome measures:-Child/young adults’ factors: The Child and Adolescent Factors Inventory (CAFI). The 15-item CAFI is a parent-report outcome measure consists of a list of problems or impairments related to the patients’ health, cognitive, physical and psychological functioning. The CAFI is also part of the CFFS. Each item is rated on a 3-point scale: 1 = No problem; 2 = Little problem; 3 = Big problem. The final score is the sum of all item ratings divided by the maximum possible score of 54 (e.g., 36/54 = 0.67). This score then was multiplied by 100 to create an outcome on a 0–100-point scale. Higher scores indicate a greater extent of problems [45].-Environmental factors: Child and Adolescent Scale of Environment (CASE): The 18-item CASE is a parent-reported outcome measure and is designed to assess the frequency and impact of environmental barriers experienced by children and young adults with disabilities. The CASE is also part of the CFFS. Similar to the CAFI, each item is rated on a 3-point scale: 1 = No problem; 2 = Little problem; 3 = Big problem and the final score is calculated in the same way. Again, higher scores indicate a greater extent of problems [45].-Health-related Quality of Life (HRQoL): The 23-item Pediatric Quality of Life InventoryTM Generic Core Scales 4.0 (PedsQL™ GCS 4.0) is a patient-report and parent-report outcome measure and is used to determine the patients’ HRQoL [49] It is available in a Dutch language version and is validated for different age ranges and diagnoses (also for the for the pediatric TBI population) [50] It yields a total-score and 4 dimension-scores i.e., physical functioning (8 items), emotional functioning (5 items), social functioning (5 items), school/work functioning (5 items) [49] Items are answered on a Likert-scale (0 = never to 4 = almost always) and thereafter linearly transformed to a 0–100 scale (0 = 100, 1 = 75, 2 = 50, 3 = 25, 4 = 0). The results, items summed and divide by the number of items answered gives a final score between 0–100, with lower scores indicating diminished HRQoL [49,51].

### 2.5. Statistical Analysis

#### 2.5.1. Characteristics

Patients’ injury, demographic and family related characteristics were described using descriptive statistics. All continuous variables were expressed as medians with interquartile ranges (IQR) and means with standard deviations (SD), based on their distributions (Kolmogorov-Smirnoff (K-S) test). Characteristics were presented for the total group and for the group of children (5–17 years) and the group of young adults (18–24 years) separately. The age categorization for children and young adults is in line with the Committee on Improving the Health, Safety, and Well-Being of Young Adults (Washington DC, 2015) and previous Dutch studies in patients with ABI [50,52,53,54].

#### 2.5.2. Primary/Secondary Outcome Measures

Regarding the primary (CASP) and secondary outcome measures (CAFI, CASE, PedsQL™ GCS-4.0), descriptive statistics were used to describe both the patient-report and the parent-report total scores of the CASP and the PedsQL™ GCS-4.0 and, if applicable, the domain scores. The CAFI and CASE were described similar as the CASP and the PedsQL™ GCS-4.0 but were only parent-report outcome measures. All outcomes were expressed as medians with IQRs (K-S test). To assess the potential correlation between the total scores of the CASP, PedsQL™ GCS-4.0 for HRQoL (patient/parent-report) and the CAFI/CASE (parent-report), Spearman correlations were calculated (Rho; ρ) and were considered: very strong, if >0.70; strong, if 0.40–0.69; moderate, if 0.30–0.39; weak, if 0.2–0.29; and negligible, if <0.19 [55].

#### 2.5.3. Four-Level Group Categorization (CASP)

To interpret how limited the patients’ participation restrictions were (patient-report and parent-report), the 4-level group categorization was used i.e., “full participation”/“somewhat limited”/“limited”/“very limited” participation. The CASP median (IQR) total scores are presented for all 4 group category levels. Per group (1 to 4), patient characteristics i.e., age, gender, time between administration to rehabilitation and ABI onset (<6 months or ≥6 months between onset and referral), cause; TBI/nTBI; and severity levels TBI; mild/moderate-severe, were reported (using descriptive statistics). Finally, within-group median (IQR) total scores of the CAFI/CASE/PedsQL™ GCS-4.0 were reported.

#### 2.5.4. Comparing Patients’ and Parents’ Perspectives

To compare outcomes, data from the patient-report and parent-report CASP versions, Wilcoxon signed-rank tests were used, for children and young adults separately. To test agreement between patients and parents additionally the Intraclass Correlation Coefficients (absolute agreement, single measures; ICC’s) were calculated both for the CASP total and CASP domain scores. ICC scores were considered poor, if <0.40; moderate, if 0.41–0.60; good, if 0.61–0.80; excellent, if >0.81 [56]. Regarding the results obtained by using the 4-level categorization system, Weighted kappa (K_w_) with linear weights was used to assess agreement between patients’ and parents’ scores [57,58]. The Strength of agreement is considered: poor, if < 0.20; fair, if 0.21–0.40; moderate, if 0.41–0.60; good, if 0.61–0.80; very good, if 0.81–1.00 [57,58,59]. A Bonferroni correction was performed to account for multiple testing (the α-value divided by the number of analyses on the dependent variable did not exceed 0.05). Outcomes were described for the total group, for children (5–17 years), and for young adults (18–24 years) separately. Descriptive statistics were used to describe the CASP median (IQR) total scores, domain scores and categorization (counts, percentages). Differences/similarities in participation restriction categorization were described as follows: patients scoring in the same category as their parents, patients scoring themselves 1 to 3 categories lower than their parents, and patients scoring themselves 1 to 2 categories higher than their parents.

All analyses were performed using SPSS 24.0 for Windows (IBM, SPSS Statistics for Windows, Version 24.0. IBM Corp, Armonk, NY, USA). The level of significance was set at *p* < 0.05 for the Spearman Rho correlation, Wilcoxon signed rank and ICC tests.

## 3. Results

### 3.1. Characteristics

Patient, family and injury related characteristics are described in Table 1. The flow of all eligible participants for the current analyses can be found in Figure 1. The data of two-hundred-sixty patients, (217 children (83%) and 43 young adults (17%)) and/or their parents was analysed. In total, there were 223 patient- and 245 parent-reported questionnaires completed and there were 209 patient-parent pairs (see Table 1 and Figure 1). One hundred and ninety-five (75%) patients had TBI of which 151 were mild TBI (77%). One hundred and thirty-five patients were female (52%). Ninety-six patients (39%) were referred to the rehabilitation center more than six months after brain injury onset. The median age of the patients in the group of children (5–17 years) was 14 years (IQR 11–16), and 18 (IQR 18–19) in the ≥18-year-old age group.

### 3.2. Participation Outcomes

Regarding participation outcomes in our population, as seen in Table 2, the median CASP total score reported by patients (*n* = 223) was 82.5 (IQR: 67.5–90.0), and by parents (*n* = 245) was 91.3 (IQR: 80.0–97.5). As seen in Table 2, Figure 2a,b, the lowest scores were found in the domain score “community participation” i.e., median patient-report score 75.0 (IQR: 56–92), median parent-report score 87.5 (IQR: 75–100). The highest median scores were found in the ‘home participation’ domain score for patients (87.5, IQR: 75–96), and in the “school/work participation” domain score for parents (95.0, IQR: 83–100).

Secondary outcome measures are also presented in Table 2. Regarding HRQoL, the median PedsQL™ GCS-4.0 patient-report total score was 65.2, (IQR: 53–78), and the median parent-report score was 60.9 (IQR: 48–75). The parent-report median scores in the CAFI (child/young adult factors) and CASE (environmental factors) were: 56.9 (IQR: 49–65) and 39.0 (IQR: 33–51), respectively. Spearman’s rho correlations between the CASP scores and the CAFI/CASE and HRQoL were significant (*p* < 0.01) and strong ranging between: ρ 0.53–0.67.

### 3.3. Four-Level CASP Categorization

Table 3 shows within-group (patient/injury-related) characteristics, and CASP/CAFI/CASE/HRQoL scores of participation restrictions (patient-report and parent-report where applicable) in our cohort, organized by the 4-level CASP participation restrictions categorization. Eighty-nine percent of the patients, and 73% of the parents reported patients’ participation restrictions in more than one CASP domain. Forty-three percent (patient-reported) and 45% (parent-reported) reported CASP total scores that fell in the “somewhat limited” category. Twenty six percent (patient-report) and 10% (parent-report) reported CASP total scores that fell in the “very limited” category. In this “very limited” category, median CASP scores were 57.9 (IQR: 50–64) for patient-report data, and 61.4 (IQR: 49–65) for parent-report data. Patients who fell in this ‘very limited’ category, had a median age of 15 years (both in the patient and parent-reported category), 45–52% were female, 64–78% had a TBI and 33–40% were referred for rehabilitation more than 6 months after ABI onset. Lower participation CASP scores, i.e., category levels up to category 4, also showed lower (diminished) patient and parent report HRQoL scores, and higher (more problems) parent report CAFI/CASE scores.

### 3.4. Differences in Patients’ and Parents’ Perspectives

In Table 4, the differences in participation outcomes between patients and parents (paired samples) is reported. Regarding the total paired-sample group (*n* = 209), there was moderate agreement in participation total CASP and domain outcomes between patients and their parents i.e., ICC = 0.42–0.57, all *p* < 0.001. In the group of children (5–17 years, *n* = 176) moderate agreement was found between patients’ and their parents’ total CASP and domain scores (ICC = 0.43–0.55, all *p* < 0.001). In the young adult (≥ 18 years, *n* = 33) group, there was poor-moderate patient/parent agreement between patient- and parent report scores on all CASP domains (ICC = 0.37–0.59, all *p* < 0.001). Regarding the categorical data on the 4-level categorization system, a fair to moderate agreement was found between the patients and parents; “moderate” in children; K_w_: 0.42 (95%CI 0.32–0.52, *p* < 0.001), and “fair” in young adults; K_w_: 0.27 (95%CI 0.08–0.46, *p* < 0.05). Regarding the differences in categorization between patients and their parents, in the total paired-sample group, 38% of the patients scored themselves in a lower CASP level category than their parents. In the group of children, the same percentage was found (38%), while in the young adult group 51% scored themselves in a lower category than their parents.

## 4. Discussion

According to data gathered before/on the first appointment for routine outpatient rehabilitation for children and young adults with ABI and their parents in multiple rehabilitation centers, 88% (patient-reported) and 73% (parent-reported) of the patients have participation restrictions that can be classified as “somewhat limited” to “very limited”, with a considerable number of patients (25, parent reported and 58, patient reported) that can be classified as “very limited”. The large majority was classified in the “somewhat limited” category. Overall, patients consistently reported more severe participation restrictions than parents. There was a greater discrepancy in the levels of participation restrictions between patients and parents in the young adult group compared to the children group.

### 4.1. Participation Restrictions

These results confirm that experiencing participation restrictions is common in pediatric patients with brain injuries (TBI/nTBI) [2,6,7,9,12,16,17,25,30,31,32,33,34,35,36,41]. Furthermore, the results we found, pointed out that the rehabilitation referred group had more participation restrictions compared to a Dutch hospital-based cohort [2]. In the current analyses of data among patients referred to an outpatient rehabilitation center, the vast majority reported participation restrictions in one or more domains of the CASP. This proportion was relatively high as compared to the 25–80% reported in a systematic review of studies on participation restrictions in children and youth with ABI including in hospital-based cohorts [7]. The current analyses found that the majority of patients was classified as “somewhat limited”. These patients could also be “at risk” regarding restricted participation. In clinical practice it could also be important to monitor the patients that score relatively better than patients with more limited participation. However, future research must confirm this hypothesis by further looking into the “somewhat limited” patients. Concerning the prevalence of participation restrictions in young adults, some differences with the literature were found. A previous rehabilitation-based study, with patient and parent-reported data that focused on patients with ABI in the age group of 14–25, reported similar participation restrictions when compared to the results of the total sample from the current analyses [9]. However, more participation restrictions were found in the young adult group in the current analyses [9]. Differences could possibly be explained by differences in age inclusion. Results suggest that young adults experience more participation restrictions than children. This could be explained by the greater appeal made on for example independence, planning and coping in this transitional age group.

### 4.2. Community Participation

For both patient-report and parent-report CASP outcomes and in all (age) groups (<18 years/>18 years/total), the lowest scores were found in the domain ‘community participation’ which includes participation related to e.g., social play/leisure activities with friends, events, sports, doing groceries, communicating with others in the neighborhood. [45,47]. Restrictions in community participation could also be related to the fact that children and young adults with ABI often have difficulties in social functioning, emotional functioning, and processing sensory stimuli (after ABI onset). These competences are needed when participating in the community [7,37]. However, other factors (e.g., environmental resources, stigma, family support, as well as time allocation), may also influence community participation [14,42].

### 4.3. Correlations with the CASP and CAFI/CASE/HRQoL

In comparison to a previous Dutch study in a hospital cohort with a higher CASP total score, the mutual correlations of the CASP with the CAFI, and CASE (parent-report), were higher in this rehabilitation-based population [2]. Regarding HRQoL, in line with previous literature participation was found to be highly correlated with HRQoL (patient-report and parent-report) [9,16,35]. These results underline the interdependence of limitations on the level of participation (CASP), child/young adult factors e.g., body functions and structures (CAFI), environmental factors (CASE) and HRQoL (PedsQL GCS-4.0). These findings also support the assumption that the CASP, PedsQL GCS-4.0, CAFI and CASE are more suitable among patients that were administered to outpatient rehabilitation (and filled out the questionnaires at admission) than in patients that were in a hospital (hospital-based).

### 4.4. Notable Results Found in the Current Rehabilitation-Based Population

Notable results were found in the current analyses among the outpatient rehabilitation-based population, which were not found in previous studies [36,41].

-Firstly, the majority of children and young adults with a mild TBI reported scores in the “very limited” category. These results suggest that the TBI population experience participation restrictions no matter the initial TBI severity. Therefore, targeting and monitoring these restrictions for all TBI severities is relevant at admission to rehabilitation treatment.-Secondly, late referral (over 6 months) to outpatient rehabilitation was common across all participation category groups based on the CASP total scores, in example; “somewhat limited participation category”–“very limited participation category”. “One-third up to 45% of the patients in the different participation categories were referred for rehabilitation more than 6 months after ABI onset. This was also common among more than one-third of the patients in the “very limited” category. Several explanations can be given for a delay in referral. Medical specialists and general practitioners could potentially underestimate (long-term) problems/restrictions of patients or simply do not recognize them and/or they may not be familiar with pediatric ABI care pathways. Parents and patients do not know what signals or problems to be alert of, may tend to choose a “wait and see” approach before seeking help and/or are not familiar with ABI support pathways [5].

These findings should be discussed with professionals in acute care to increase awareness of possible consequences of later rehabilitation referral and to ultimately improve referral policies and procedures.

### 4.5. Differences in Perspectives

Regarding patients’ and parents’ perspectives, moderate agreement between patient and parent reported CASP (total and domain) scores were found. Previous studies underlined the importance of measuring both patients’ and parents’ perspectives to assess outcomes [20,36]. One Dutch study regarding adolescents and young adults with ABI found a difference between the patients and the parents CASP total score outcomes, similar to what we found in the results of the analyses [9]. Parents tend to report less participation restrictions for their children than the patients themselves, which is in contrast to previous studies with other outcomes (e.g., HRQoL; where parents usually report lower scores than their children) [9,16,17,25,30,31,32,33,35,40]. This was also found in our analyses. A large part of the patients in our cohort scored themselves in another CASP level category than their parents did. These discrepancies in reporting outcomes may be explained due to the fact that most participation activities (of the children and young adults) occur outside of the home environment where parents are not present and also, young adults spend more time away from parents than children. Our results suggest that assessing both patients’ and parents’ perspectives is important in order to identify differences and similarities. By using both perspectives, a broader view on overall functioning is attained, providing health care professionals the opportunity to consider both patients’ and parents’ perspectives when collaborating on rehabilitation goals, and make sure parents play an active role in today’s often proposed family-centered care [14].

### 4.6. Categorization of Severity of Participation Restrictions

In the currently analyzed data, a 4-level categorization was created that correspond to specific CASP score ranges to reflect the overall degree of participation restriction. This categorization was based on previously identified levels of participation suggested by one of the authors (G.B.). To date, CASP outcomes were described as just a score between 0 and 100 (lower score = more participation restrictions). To facilitate a better interpretation of the score in clinical practice, we proposed a categorization of the total score into four levels. This 4-level categorization can be used next to the original 0–100 score) to compare and report CASP outcomes. The use of cut-off values may help to contextualize and differentiate the scores for clinical practice (i.e., indication for rehabilitation, evaluation of intervention) and research. All statistical comparisons of patients’ and parents’ scores in the present study consistently demonstrated a considerable discrepancy. Poor agreement was also seen using the proposed 4-level categorization, substantiating the validity of that division. Regarding the 4 categories, the majority of the patients and their parents reported CASP scores in the ‘somewhat limited’, the ‘limited’ and ‘very limited’ categories. A quarter of the children and almost one-third of the young adults scored in the most restricted, i.e., “very limited” category. Parent and patient-report scores differed in participation restriction category in almost half of the of cases, with parent scores and categories demonstrating lower levels of participation restriction as previously described. Future longitudinal studies could use this new categorization to further evaluate its utilization, and/or to investigate recovery outcomes over time (e.g., moving to higher category level of participation) during rehabilitation treatment related to interventions.

### 4.7. Limitations

Describing analyses and results among rehabilitation referred patients resulted in a number of limitations. First, there was a relatively small sample of young adults compared to the sample of children (43 vs. 217). The explanation is merely organizational: most rehabilitation centers have a separate pediatric (<16 or <18 years) and adult (≥18 years) department where only the pediatric department was involved. Only two centers had a separate department for young adults (18–25 years) and included young adults. However, the number of included young adults was large enough to analyze and report outcomes for separately. Since, due to age and life phase, in the young adult group is a different group of patients it is recommended to include this group of patients in transition fully in future pediatric studies. Secondly, not for all patients paired sample data was available, making the analysis for the differences/similarities between patients’ and parents’ perspectives only possible for a portion of our analyses (*n* = 209). However, since we had paired sample data available for the majority of patients, we believe that outcomes are generalizable. Third, the CASP is known to have a ceiling effect [17,47] Nonetheless, in contrast to other studies reporting ceiling effects in children and young adults with ABI, these were less evident in the present analyses making the CASP a more suitable instrument for use in rehabilitation cohorts (versus patients that are hospitalized) of patients with ABI [2,17,20,47]. Furthermore, an alternative instrument that also focusses on the ABI population is lacking [17]. Finally, results of patient/parent rated outcome measures could be biased, i.e., by limitations in motivation or patients’ and/or parents’ moment bound ’stress and mood’.

### 4.8. Directions for Future Research

Interesting follow up projects could be longitudinal studies monitoring participation over time and evaluative studies using the CASP to explore the effect of rehabilitation programs for children and young adults with ABI and their families, since optimizing participation is an important rehabilitation goal. In these studies, the newly developed categorization of participation outcomes could be used and further investigated on its usefulness and robustness. Future studies should include the search for the best available participation outcome measures particularly given the number of promising participation-focused, multi-setting interventions that recently have been developed to improve participation outcomes for individual children, youth, and families [21,22,23,24]. The next challenge is to drive implementation of participation-based interventions on a larger scale, and research should be focused on enabling strategies and on cost-effectiveness of these interventions. The CASP and our newly proposed categorization of participation restrictions could support this process. 

## 5. Conclusions

A substantial portion of patients (ages 5–24 years) with acquired brain injury referred to an outpatient rehabilitation center in The Netherlands had “limited” to “very-limited” participation. Patients reported greater participation restrictions than their parents and disparities between patient reported and parent reported participation restrictions were greater in young adults than in children. Furthermore, a strong correlation was found between patient and environmental factors (CAFI and CASE), HRQoL (PedsQL GCS-4.0), and participation (CASP). Most restrictions were found in the ‘community participation’ domain. A large part of the patients with a late referral (>6 months) to rehabilitation after ABI onset reported “very limited” participation. Early referral is important as this may reduce participation restrictions. Taking into account both patients’ as well as parents’ perspectives is important in outpatient rehabilitation treatment in order to guide both patients and their parents appropriately during treatment. Furthermore, the categorization of CASP scores into 4 categories might be useful for clinical practice and research, but more study is needed to understand how this can be applied and inform participation focused clinical and practical decisions.

## Figures and Tables

**Figure 1 ijerph-18-01625-f001:**
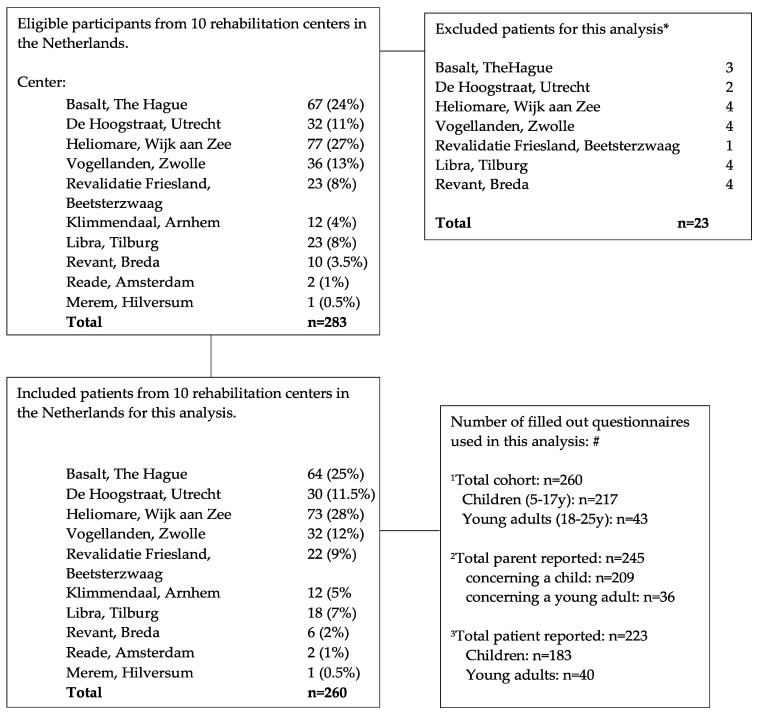
Flow of children and young adults with ABI admitted for rehabilitation and eligible for the present analysis. * Missing participants: *n* = 11 no official ABI diagnosis, *n* = 12 incomplete questionnaires. # Number of filled out questionnaires used in this analysis (total/patient-reported/parent-reported): ^1^; number of questionnaires filled out by the patient, the parents or both in total and per age group (children, adolescents and young adults). ^2^; number of questionnaires filled out by parents only in total and per age group (children, adolescents and young adults). ^3^; number of questionnaires filled out by patients only (self-reported) in total and per age group (children, adolescents and young adults).

**Figure 2 ijerph-18-01625-f002:**
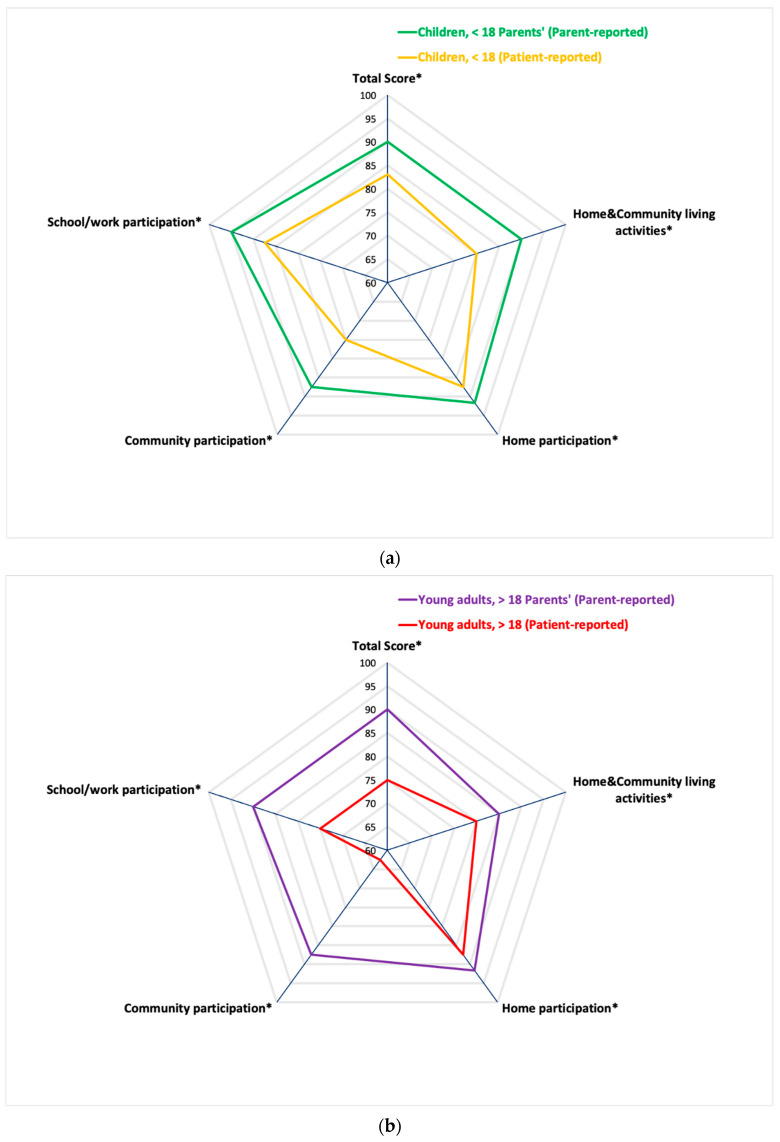
(**a**) Differences in CASP scores between Patients and Parents in children (5–17 years) with ABI. * CASP: Child and Adolescent Scale of Participation, 0–100 with lower scores indicating more participation restrictions. (**b**) Differences in CASP scores between Patients and Parents in young adults (18–24 years) with ABI. * CASP: Child and Adolescent Scale of Participation, 0–100 with lower scores indicating more participation restrictions.

**Table 1 ijerph-18-01625-t001:** Patient, family and injury characteristics of children and young adults with acquired brain injury (ABI) referred to an outpatient rehabilitation center.

Patient Injury and Demographic Related Characteristics	Children 5–17 y, *n* = 217	Young Adults ≥18 y, *n* = 43	Total Cohort 5–24 y, *n* = 260
Gender: Female *n* (%)	112 (52%)	23 (54%)	135 (52%)
Age (years) at admission			
median (IQR)	14 (11–16)	18 (18–20)	14 (11–16)
Time (months) between ABI onset and referral to rehabilitation			
median (IQR)	4.0 (1–18)	4 (2–19)	4 (1–18)
>6 months *n* (%)	87 (40%)	17 (40%)	104 (40%)
Traumatic brain injury (TBI) *n* (%)	160 (74%)	35 (81%)	195 (75%)
Severity levels TBI * *n* (%)			
Mild	124 (78%)	27 (77%)	151 (77%)
Moderate-severe	15 (9%)	5 (14%)	20 (10%)
Unknown	21 (13%)	3 (9%)	24 (13%)
Non-traumatic brain injury (nTBI) *n* (%)	57 (26%)	8 (19%)	65 (25%)
Causes nTBI *n* (%)			
Tumor	25 (44%)	2 (25%)	27 (41%)
Stroke	11 (19%)	5 (63%)	16 (25%)
Encephalitis/meningitis	10 (17%)	1 (12%)	11 (17%)
Hypoxia/intoxication	2 (4%)	0 (0%)	2 (3%)
Other/unknown	9 (16%)	0 (0%)	9 (14%)
**Family Related Characteristics**	**Children 5–17 y, *n* = 209**	**Young adults** **≥** **18 y, *n* = 36**	**Total Cohort 5–24 y, *n* = 245**
Living in a single-parent household *n* (%)	34 (16%)	8 (22%)	42 (17%)
Cultural background parents: non-Dutch *n* (%)	16 (8%)	2 (6%)	18 (7%)
Educational level parent** number (%)			
Low	7 (3%)	3 (8%)	10 (4%)
Intermediate	41 (20%)	6 17%)	47 (19%)
High	162 (77%)	27 (75%)	188 (77%)

* Based on Glasgow Coma Scale at hospital admission: “mild”—13–15, “moderate”—9–12, “severe” < 8 ** Educational level parent: low—prevocational practical education or less, intermediate—prevocational theoretical education and upper secondary vocational education, high—secondary education, higher education and/or university level education.

**Table 2 ijerph-18-01625-t002:** Total and domain scores on the CASP, CAFI, CASE and PedsQL™ GCS-4.0 (HRQoL) of children and young adults with acquired brain injury (ABI) and mutual correlations.

Outcome Measure	Domain Scores/Total Scores	Patient Report	Parent Report
		*n* = 223	*n* = 245
		Median (IQR)	Median (IQR)
CASP ^1^	Total Score	82.5 (68–90)	91.3 (80–98)
	Home/community living activities	80.0 (63–90)	90.0 (75–100)
	Home participation	87.5 (75–96)	91.7 (83–100)
	Community participation	75.0 (56–92)	87.5 (75–100)
	School/work participation	85.0 (67–95)	95.0 (83–100)
PedsQL™ GCS-4.0 (HRQoL) ^2^	Total score	65.2 (53–78)	60.9 (48–75)
	Physical health	68.8 (50–86)	68.8 (47–81)
	Emotional functioning	65.0 (45–85)	60.0 (40–75)
	Social functioning	80.0 (65–90)	75.0 (60–95)
	School/work functioning	50.0 (35–65)	50.0 (30–60)
CAFI ^3^	Total Score	NA	56.9 (49–65)
CASE ^3^	Total Score	NA	39.0 (33–51)
**Correlations ^$^**		**Patient Report**	**Parent Report**
		***n* = 223**	***n*** **= 245**
		**Rho**	**Rho**
CASP total score	HRQoL total score	ρ 0.67 **	ρ 0.62 **
CASP total score	CAFI total score	NA	ρ 0.60 **
CASP total score	CASE total score	NA	ρ 0.53 **

^1^ CASP: Child and Adolescent Scale of Participation, 0–100 with lower scores indicating more participation restrictions. ^2^ PedsQL™ Generic Core Scales 4.0 for Health-related quality of life (HRQoL): 0–100 with lower scores indicating lower HRQoL. ^3^ CAFI: Child and Adolescent Factors Inventory (CAFI), and CASE: Child and Adolescent Scale of Environment, 0–100 with higher scores indicating more problems. ^$^ ρ = Spearman’s rho (ρ) correlation. ** *p* < 0.001.

**Table 3 ijerph-18-01625-t003:** Within group characteristics of children and young adults with acquired brain injury (ABI) based on CASP participation restriction categorization.

Patient-Report (CASP) *n* = 223 (100%)										
Category	*n* (%)	CASP Totalscore ^#^ Median (IQR)	Age: Median (IQR)	Gender Female: *n* (%)	^$^ Admin to Rehab ≥6 m: *n* (%)	Cause TBI: *n* (%)	Severity TBI Mild: *n* (%)	^a^ HRQoL Median (IQR)		
^1^ Full participation	25 (11%)	98.7 (98–99)	15 (12–16)	12 (48%)	13 (52%)	18 (72%)	14 (88%)	80.4 (75–86)		
^2^ Somewhat limited participation	95 (43%)	86.8 (84–91)	14 (12–16)	50 (53%)	35 (37%)	72 (76%)	55 (86%)	75 (63–82)		
^3^ Limited participation	45 (20%)	75 (71–78)	15 (11–17)	29 (64%)	15 (33%)	32 (71%)	24 (83%)	62 (52–68)		
^4^ Very limited participation	58 (26%)	57.9 (50–64)	15 (13–16)	26 (45%)	15 (33%)	45 (78%)	39 (97%)	47.3 (38–58)		
**Parent-Report (CASP) *n* = 245 (100%)**										
**Category**	***n*** **(%)**	**CASP** **Totalscore ^#^** **Median (IQR)**	**Age:** **Median (IQR)**	**Gender:** **Female: *n* (%)**	**^$^ Admin** **to Rehab ≥6 m *n* (%)**	**Cause:** **TBI**	**Severity TBI: Mild**	**^a^ HRQoL** **Median (IQR)**	**^b^ CAFI** **Median (IQR)**	**^b^ CASE** **Median (IQR)**
^1^ Full participation	67 (27%)	100 (98–100)	15 (11–16)	33 (49%)	24 (36%)	58 (87%)	45 (92%)	75 (63–83)	47.1 (43–57)	34.4 (33–36)
^2^ Somewhat limited participation	111 (45%)	91.3 (88–94)	13 (10–15)	59 (53%)	43 (39%)	79 (71%)	62 (86%)	64.1 (54–77)	56.9 (51–63)	39.8 (33–50)
^3^ Limited participation	42 (17%)	76.3 (73–80)	14 (10–16)	23 (55%)	19 (45%)	28 (67%)	24 (89%)	47.8 (41–55)	63.7 (55–71)	48.8 (43–62)
^4^ Very limited participation	25 (10%)	61.4 (49–65)	15 (12–17)	13 (52%)	10 (40%)	16 (64%)	12 (92%)	42.4 (35–48)	64.7 (59–74)	53.2 (41–67)

^#^ CASP: Child and Adolescent Scale of Participation, 0–100 with lower scores indicating more participation restrictions. Categories: ^1^ Full participation: Group 1, Between 97.5–100: Participating in activities the same as or more than other peers, ^2^ Somewhat limited participation: Group 2, Between 97.5–81: Participating in activities somewhat less than other peers, ^3^ Limited participation: Group 3: Between 81–68.5: Participating in activities less than other peers, ^4^ Very limited participation: Group 4, Below (<) 68.5: Participating in activities much less than other peers. [45] ^$^ Time (months) between administration to rehabilitation and ABI onset (more than 6 months). ^a^ Patient and parent-report PedsQL™ Generic Core Scales 4.0 for health-related quality of life (HRQoL), 0–100 with lower scores indicating lower HRQoL. ^b^ Parent-report CAFI: Child and Adolescent Factors Inventory, and Parent-report CASE: Child and Adolescent Scale of Environment, 0–100 with higher scores indicating more problems.

**Table 4 ijerph-18-01625-t004:** Differences and similarities between patient and parent CASP participation scores and categories.

Paired Samples Total Group (5–24 Years) *n* = 209				
CASP	Patient Report Median (IQR)	Parent Report Median (IQR)	Wilcoxon Z ^#^	ICC ^$^
Total Score	82.5 (68–90)	90.0 (80–97)	−8.2 **	0.54
Home/community living activities	80.0 (63–90)	90.0 (75–100)	−5.9 **	0.51
Home participation	87.5 (75–96)	91.7 (83–100)	−5.9 **	0.42
Community participation	75.0 (56–92)	87.5 (75–100)	−8.5 **	0.51
School/work participation	85.0 (66–90)	95.0 (80–100)	−6.2 **	0.57
**CASP Categorization**	**Patient report** **Number (%)**	**Parent Report** **Number (%)**	**Patient/Parent** **Categorization ^**	**Number (%)**
			Same as parents	110 (53%)
- Full	23 (11%)	51 (24%)	Different from parents	99 (47%)
- Somewhat limited	92 (44%)	98 (47%)	a:1 category worse	54 (26%)
- Limited	41 (20%)	37 (18%)	b: 2 categories worse	15 (7%)
- Very limited	53 (25%)	23 (11%)	c: 3 categories worse	10 (5%)
	K_w_: 0.40 (95%CI 0.31–0.49), *p* < 0.001		d: 1 category better	18 (9%)
			e: 2 categories better	2 (1%)
**Paired Samples Children (5–17 Years) *n* = 176**				
**CASP**	**Patient Report** **Median (IQR)**	**Parent Report Median (IQR)**	**Wilcoxon** **Z ^#^**	**ICC ^$^**
Total Score	83.1 (69–90)	90.0 (80–97)	−7.4 **	0.54
Home/community living activities	80.0 (63–90)	90.0 (75–100)	−5.2 **	0.51
Home participation	87.5 (75–96)	91.7 (83–100)	−5.4 **	0.43
Community participation	75.0 (56–92)	87.5 (75–100)	−7.4 **	0.52
School/work participation	87.5 (70–96)	95.0 (82–100)	−5.6 **	0.55
**CASP Categorization**	**Patient Report** **Number (%)**	**Parent Report** **Number (%)**	**Patient/Parent** **Categorization ^**	**Number (%)**
			Same as parents	99 (53%)
- Full	20 (11%)	41 (23%)	Different from parents	77 (47%)
- Somewhat limited	83 (47%)	86 (49%)	a: 1 category worse	42 (24%)
- Limited	30 (17%)	31 (18%)	b: 2 categories worse	11 (6%)
- Very limited	43 (24%)	18 (10%)	c: 3 categories worse	8 (5%)
	K_w_: 0.42 (95%CI 0.32–0.52), *p* < 0.001		d: 1 category better	14 (8%)
			e: 2 categories better	2 (1%)
**Paired Samples Young Adults (18–24 Years) *n* = 33**				
**CASP**	**Patient report** **Median (IQR)**	**Parent report** **Median (IQR)**	**Wilcoxon** **Z ^#^**	**ICC ^$^**
Total Score	75.0 (65–86)	90.0 (78–99)	−3.6 **	0.56
Home/community living activities	80.0 (66–90)	85.0 (75–100)	−2.8 *	0.52
Home participation	87.5 (75–90)	91.7 (79–100)	−2.3 *	0.37
Community participation	62.5 (50–84)	87.5 (75–100)	−4.0 *	0.48
School/work participation	75.0 (55–90)	90.0 (74–100)	−2.8 *	0.59
**CASP Categorization**	**Patient report** **Number (%)**	**Parent Report** **Number (%)**	**Patient/Parent Categorization ^**	**Number (%)**
			Same as parents	12 (37%)
- Full	3 (9%)	10 (30%)	Different from parents	21 (63%)
- Somewhat limited	9 (27%)	12 (36%)	a:1 category worse	11 (33%)
- Limited	11 (33%)	6 (18%)	b: 2 categories worse	4 (12%)
- Very limited	10 (30%)	5 (15%)	c: 3 categories worse	2 (6%)
	K_w_: 0.27 (0.08–0.46), *p* < 0.05		d: 1 category better	4 (12%)
			e: 2 categories better	0 (0%)

^1^ CASP: Child and Adolescent Scale of Participation, 0–100 with lower scores indicating more participation restrictions. ^#^ Z scores for Wilcoxon signed-rank test for nonparametric data outcomes * *p* < 0.05, ** *p* < 0.001; ^$^ ICC; Intraclass Correlation Coefficients rated: <0.40: poor; 0.41–0.60: moderate; 0.61–0.80 good; >0.81: excellent. K_w_: Weighted Kappa interpretation (categorical CASP score): <0.20: poor, 0.21–0.40: fair, 0.41–0.60: moderate, 0.61–0.80: good, 0.81–1.00: very good—agreement. ^ Patient categorization compared to parents’ categorization: The differences in categorized participation between patients and their parents, a: Patients that scored 1 category worse than their parents, b: Patients that scored 2 categories lower than their parents, c: Patients that scored 3 categories lower than their parents, d: Patients that scored 1 category better than their parents, e: Patients that scored 2 categories better than their parents.

## Data Availability

Data used in this study is stored in a central database at Basalt Rehabilitation center, The Hague in the office of innovation, quality and research and can be available when requested.

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
