# Peer review of "Participation Restrictions among Children and Young Adults with Acquired Brain Injury in a Pediatric Outpatient Rehabilitation Cohort: The Patients’ and Parents’ Perspective"

_ijerph, 2021, doi:10.3390/ijerph18041625_

Round 1

Reviewer 1 Report

Thanks for inviting me to review this manuscript that investigated participation restrictions in children and young adults with ABI from patients’ and parents’ perspectives. Participation restriction is an important issue and has been rarely investigated, especially through the use of the categorization approach. Most studies investigated participation restrictions of children with disabilities in comparison with their typically developing peers and, if children with disabilities exhibited lower participation patterns, they would be considerate as having restricted participation. The current manuscript took a step further by proposing a new four-category system of the CASP specifically in children and young adults with ABI. The findings of this manuscript would contribute to child participation research considerably. I also echo the authors’ viewpoint that this new categorization system could be useful for clinical practice and epidemiological studies. The following comments are provided to further polish the flow and analytic approach of this study.

  1. 1. In this study, the authors attempted to investigate the factors associated with participation restrictions. This was done by examining the relationship between the CASP scores/categories and other secondary measures including CAFI, CASE, and PedsQL as well as disability severity and child age. While this agenda was stated a bit in Introduction section (line 78-84), it was not included as the study aim in the following paragraph, making it irrelevant to the entire topic (i.e., the difference between patients and parent’s perspectives). It would be good to state clearly if this is also included as one of research aims/questions.
  2. 2. Line 152-163, the four-category system of the CASP was made in the study, based on the levels of participation categories created by Prof. Gary Bedell in two of his publications. This statement could be misleading, as people may think that the system was proposed by Prof. Bedell in his study but, in fact, the system was newly proposed by the authors in the present study. I would suggest the authors to describe more about how the four-category system was developed from the original 12-category system developed by Prof. Bedell. Also were there any validation based on content experts who assisted in the decision of the cut-off scores of each category? This information is important, because the finding of this study recommend the system to be used in clinical practice. However, the content validity evidence has to be presented as well.
  3. 3. Line 221-233, Wilcoxon signed-rank test and ICC were used to examine the difference and consistency of CASP scores between patients’ and parents’ perspectives. This part is fine for me, but it would be good to specify that the outcome was the CASP total and domain scores (not the categorization outcome). On the other hand, there is a statistic the authors may consider to analyze the agreement between the categorical results between patients and parents. It is weighted kappa, commonly used to examine the agreement between categorical data.
  4. 4. Line 340-341, it seems to me that the authors used the categories “limited” and “very limited” to define participation restrictions experienced by children and young adults with ABI. How about those patients being categorized in “somewhat limited” level? Would the authors consider those patients to have no restricted participation or to be at risk of restricted participation? This could be discussed as the categorization is relatively new to the audience.
  5. 5. Lines 346-371, in fact, community participation is always lowest compared to other areas of participation in either children with or without disabilities. This may not be only related to the functional difficulties in children and youth with ABI. Other factors such as environmental resources, stigma, family support, as well as time allocation could also lead to relatively low participation in community activities for children and youth.
  6. 6. In Conclusion section, if investigating the factors associated with participation restrictions is included as one study aim, the finding could be summarized here for audiences.
  7. 7. Your Table 3 is out of the format. It was difficult for me to identify the results between the patients and parents.
  8. 8. Line 18, the sentence “the differences in perspectives between … is scarce” seems incomplete.

Reviewer 2 Report

This is an interesting well-referenced study; I think it could be improved by attention to the following comments:

  1. The English is very good, there is one spelling mistake on line 64 'which' not 'witch'. Line457 paired sample data were not available for all patients. 
  2. I had to read the paper a few times to be clear that the study started prior to referral to an outpatient rehabilitation facility. Were the children referred on the basis of continuing problems or would any child be referred 6 months after the ABI? A related question - had the participants received any rehabilitation prior to entry to the study?
  3. Parents are likely to be more influential on participationwith younger children. It might have been useful to divide the young group into younger/older children. Also community involvement may have been different in younger and older children
  4. I would like to see the rationale for usung non-parametric statistics. As presented you have a lot of wilcoxon comparisons. Did you use any correction to significance levels for multiple tests? Looking at your results it seemed that the means may not have been very different from the medians. It would be helpful to provide means. Clearly using parametric statistics would have been more powerful and would have allowed for consideration of the effects of gender and family characteristics on the results.
  5. My major query regards the rationale for the study. I assume that these are baseline studies and that the proposal is to carry out a 2nd round of tests after the outpatient rehabilitation treatment (altho' one might expect some improvement with time anyway). A more important and substantial paper could have contained both of these results. 

Round 2

Reviewer 2 Report

The changes clarify some of the issues that I had with the paper

I hope that parametric statistics will be used in the follow-up, as they are much more powerful.

Also I think not analysing gender and family characteristics in relation to participation is a pity if only to confirm previous studies